# Intratumoral Flow Void Diameter as a Predictor of High Intraoperative Blood Loss in Palliative Excisional Surgery for Metastatic Spinal Tumors

**DOI:** 10.3390/cancers16244124

**Published:** 2024-12-10

**Authors:** Yuji Ishino, Satoshi Kato, Noriaki Yokogawa, Takaki Shimizu, Masafumi Kawai, Takaaki Uto, Kazuhiro Nanpo, Megumu Kawai, Satoru Demura

**Affiliations:** Department of Orthopaedic Surgery, Graduate School of Medical Sciences, Kanazawa University, Kanazawa 920-8641, Japan; tr5xb45kyzw@gmail.com (Y.I.); chakkun1981chakkun@yahoo.co.jp (N.Y.); takaki.shimizu0928@gmail.com (T.S.); peru224@gmail.com (M.K.); 10353utta@gmail.com (T.U.); k.nanpo.3500@gmail.com (K.N.); mkawai199502@gmail.com (M.K.); msdemura@gmail.com (S.D.)

**Keywords:** intratumoral flow void, spinal metastatic tumor surgery, intraoperative blood loss

## Abstract

Metastatic spinal tumors can lead to many perioperative complications, particularly massive intraoperative blood loss (IBL). This study investigated whether intratumoral flow void (IFV) on standard magnetic resonance imaging (MRI) is associated with IBL during palliative excisional surgery for metastatic spinal tumors. By reviewing 88 previous surgical cases, multiple regression analysis revealed that IFV diameter was an independent predictor for IBL in metastatic spinal tumor surgery. A large IFV diameter indicates a hypervascular tumor with intratumoral blood vessels, which makes it challenging to control hemorrhage during surgery, making this novel marker useful for identifying patients at a higher risk of significant bleeding. Therefore, preoperative evaluation of IFV diameter can help spine surgeons manage hemorrhage risk, leading to improved surgical outcomes in patients with metastatic spinal tumors. This study underscores the need for future research to validate the utility of IFV diameter as a vascular assessment tool for metastatic spinal tumors.

## 1. Introduction

The spine is a common site of metastasis, which leads to neurological dysfunction, reduced mobility, and intense pain [1,2]. Managing vertebral metastases requires individualized treatment that typically involves a multidisciplinary approach [3]. Advances in surgical techniques and instruments, including intralesional debulking and en bloc resection, have enabled more aggressive interventions aimed at improving local disease control; however, massive intraoperative blood loss (IBL) remains a major complication of metastatic spinal tumor surgery (MSTS) [4,5]. A previous meta-analysis reported pooled blood loss of 2180 mL in MSTS [6].

Several factors, including hypervascular tumors, surgical site magnitude, and surgical duration, can influence blood loss in MSTS [7,8,9]. Despite extensive research, no consensus on the typical volume of blood loss during these surgeries exists, indicating a need for further investigation into these specific risk factors. Hypervascular tumors, which are rich in intratumoral blood vessels, require preoperative embolization to reduce blood loss in MSTS [10,11]. This indicates that hemorrhage from intratumoral blood vessels may significantly contribute to IBL. Consequently, preoperative radiological assessment of intratumoral blood vessels is essential in MSTS, particularly in the context of tumor excision.

The intratumoral flow void (IFV) has been reported as a finding indicative of intratumoral blood vessels on standard MRI of metastatic bone tumors [12,13,14]. IFV results from signal loss owing to rapid blood flow or turbulence-related dephasing. The flow voids appear as serpiginous tubular areas devoid of signals on all pulse sequences [15]. A few studies have primarily focused on the presence of IFV to assess vascularity in metastatic spinal tumors [16,17]; however, no studies have specifically addressed the morphological characteristics of the IFV or the association between IFV and IBL in MSTS. Therefore, this study aimed to investigate the association between IFV diameter and IBL in palliative excisional surgery for metastatic spinal tumors.

## 2. Materials and Methods

### 2.1. Patient Population

We retrospectively analyzed patients who underwent MSTS at our institution between January 2010 and March 2024. Surgical indications included neurological deficits, spinal instability, intractable pain due to spinal tumors, or a combination of these factors. The inclusion criteria were diagnosis of metastatic spinal tumors, palliative excisional surgery, and availability of preoperative MRI. The exclusion criteria were the absence of tumor excision, preoperative embolization, and incomplete medical records. Approval from the hospital’s institutional review board (IRB) was obtained before data collection. The requirement of individual patient consent was waived owing to the retrospective design of the study.

### 2.2. Data Collection

The data on the demographic, oncological, surgical, and radiological characteristics of the patients were retrospectively collected. The collected variables included age, sex, body mass index (BMI), tumor location, histological tumor type, spinal instability neoplastic score (SINS) [18], revised Tokuhashi score [19], preoperative radiation, presence of IFV, IFV diameter, number of resected and instrumented vertebrae, surgical duration, and IBL.

The histological tumor types were divided into hypervascular, including renal cell carcinoma (RCC), hepatocellular carcinoma (HCC), and thyroid carcinoma, and non-hypervascular tumor types [20,21]. The radiological evaluation focused on the presence of IFV and the measurement of IFV diameter. MRI of the spine was performed within 2 weeks before surgery, and the images were available for review. The imaging protocol included T1-weighted fast spin-echo, T2-weighted fast spin-echo, and short-tau inversion recovery (STIR) sequences, which were acquired using a 3T whole-body scanner with a slice thickness of 3–4 mm. Intravenous gadolinium was not administered routinely. The MRI scans were obtained in the sagittal, axial, and coronal planes using these sequences. The IFV diameter was defined as the maximum vertical distance across the tubular structure (Figure 1). The presence of the IFV was assessed by two spine surgeons with expertise in spinal imaging. To minimize potential bias, both surgeons independently evaluated the MRI scans in a blinded manner without access to clinical or demographic information. Discrepancies in the evaluation of the presence of IFV were resolved through reassessment and consensus between the two surgeons. For cases in which the IFV was absent, the IFV diameter was recorded as 0 mm. The IFV diameter was independently measured by each surgeon, and the mean value was used to enhance the measurement reliability. The intraclass correlation coefficient (ICC) was used to assess the reproducibility of the IFV diameter measurements. All measurements were performed using standardized procedures and digital tools to ensure precision and consistency.

### 2.3. Statistical Analysis

All statistical analyses were performed using SPSS version 29 (IBM Corp., Armonk, NY, USA). Categorical variables were presented as numbers and percentages, whereas continuous variables were presented as mean ± standard deviation or median and range, depending on the distribution of data. The median and interquartile range (IQR) were reported for continuous variables with non-normal distribution. The relationship between these factors and IBL was examined using univariate and multivariate regression analyses. Statistical significance was defined as *p* < 0.05.

## 3. Results

### 3.1. Demographic Characteristics

A total of 119 patients who underwent palliative surgery for metastatic spinal tumors at our institution were assessed for eligibility. The exclusion criteria were met by 21 patients with no tumor excision, 5 patients with preoperative embolization, and 5 patients with incomplete surgical parameters. Ultimately, 88 patients were included in the study, comprising 56 (63.6%) men. The mean age was 61.6 ± 13.7 years, with a range of 13–85 years. Hypervascular tumor type was identified in 19 patients (21.6%), including seven cases of RCC, eight cases of HCC, and four cases of thyroid carcinoma. Non-hypervascular tumor types were identified in 69 patients (78.4%), including 14 cases of lung cancer, 10 cases of breast cancer, and 7 cases of prostate cancer. The overall mean IBL was 420 ± 387 mL with a range of 40–2150 mL. The baseline and clinical characteristics of patients are summarized in Table 1.

### 3.2. Intratumoral Flow Void

The IFV was identified in 71 patients (80.7%). The mean IFV diameter was 1.8 ± 1.3 mm, with a range of 0–4.3 mm. The ICC indicated high agreement between the measurements of the IFV diameter by the two spine surgeons (ICC = 0.85, 95% confidence interval [CI]: 0.80–0.89). The amount of IBL increased progressively when the IFV diameter was segmented at 1 mm intervals (Figure 2).

### 3.3. Related Factors Associated with IBL

The univariate analysis identified the hypervascular tumor type, number of instrumented vertebrae, surgical duration, presence of IFV, and IFV diameter as significant factors associated with IBL (Table 2). Variables with a *p*-value < 0.05 in the univariate analysis were included in the multivariate analysis. IFV was excluded from the multivariate analysis since IFV diameter inherently includes information about the presence of IFV.

In the multivariate analysis, IFV diameter (unstandardized coefficient: 171, *p* < 0.01) and surgical duration (unstandardized coefficient: 1.60, *p* < 0.01) were identified as independent factors significantly associated with IBL. The standardized coefficients for IFV diameter and surgical duration were 0.56 and 0.27, respectively (Table 3). The regression model showed a significant fit, with an adjusted R² of 0.48 (*p* < 0.01). Furthermore, no multicollinearity concerns were observed, as all variance inflation factor values were below 5.

## 4. Discussion

This is the first study to reveal an association between IFV diameter and IBL during palliative excisional surgery for metastatic spinal tumors. The univariate analysis showed that when the IFV diameter was larger, more blood loss occurred during surgery. Even after considering the histological tumor type and surgical site magnitude, IFV diameter remained an independent predictor in the multivariate analysis. Our findings suggest that a large IFV diameter indicates a hypervascular tumor with intratumoral blood vessels that are difficult to control when bleeding occurs during the resection of metastatic spinal tumors.

Previous studies on MRI vascularity assessment defined hypervascularity based solely on the presence of flow void, intralesional hemorrhage, feeding vessels, or enhancement on contrast-enhanced MRI. These criteria are not considered reliable indicators for evaluating the vascularity of metastatic spinal tumors [16,17,22]. In contrast, given the positive correlation observed between IFV diameter and IBL, we propose that IFV diameter could serve as a new quantitative marker for evaluating tumor vascularity. This possibility can be understood through Poiseuille’s law, a well-known physical principle that posits that the flow rate *Q* through a vessel is proportional to the fourth power of its radius *r* [23]. Evaluation of the IFV diameter could reflect the overall vascularity of a metastatic spinal tumor with the presence of rapidly flowing blood vessels within the tumor.

Radiological vascularity assessment of metastatic spinal tumors relies on digital subtraction angiography (DSA), which is considered the gold standard for evaluating tumor vascularity [11,17]. DSA is both invasive and resource-intensive and requires specialized expertise from radiologists, which limits its practicality in routine clinical settings. The DSA grade was classified based on the degree of enhancement compared to normal vertebrae, the presence of capillaries, and the presence of shunts [24,25]. The DSA grade incorporates indicators that reflect the size of the vessels within the tumor. Therefore, in this study, we evaluated the IFV diameter, a marker of intratumoral vessels, using standard MRI as a simple and non-invasive method. This new approach could allow spine surgeons to better plan surgeries, including the need for preoperative embolization, blood transfusion, or hemostatic agents. The IFV diameter on standard MRI would be particularly a useful finding in cases where DSA is not feasible.

Previous studies have demonstrated the potential utility of non-invasive imaging techniques, such as dynamic contrast-enhanced magnetic resonance imaging (DCE-MRI), time-resolved contrast-enhanced magnetic resonance angiography (TR-CE-MRA), and computed tomography digital subtraction angiography (CT-DSA), for evaluating vascularity in spinal metastases. DCE-MRI offers the advantage of providing quantitative perfusion parameters without the risk of radiation exposure, allowing for a clear distinction between hypervascular and hypovascular lesions [26,27,28,29]. TR-CE-MRA sequences are widely used to detect arteriovenous shunts in the spinal and paraspinal regions [30]. Premat et al. reported that TR-CE-MRA can reliably differentiate hypervascular lesions from non-hypervascular lesions in spinal metastases with a sensitivity of 97.9%, specificity of 71.4%, and overall accuracy of 94.6% [31]. CT-DSA allows the acquisition of multiple time phases after contrast administration, enabling reliable quantitative and qualitative evaluation of spinal metastasis vascularity [32]. However, these techniques exhibit several limitations. First, their high cost and technical complexity hinder widespread use in clinical practice. In addition, some of these methods require the use of nephrotoxic contrast agents, which pose a significant burden on patients. Furthermore, radiation exposure in dual-energy CT angiography and the need for additional post-processing in dynamic contrast-enhanced MR perfusion sequences are limiting factors for routine clinical applications. Conversely, IFV diameter, which can be assessed using standard MRI not only by radiologists but also by spinal surgeons, could be a widely applicable and practical vascularity assessment method in general healthcare settings.

In addition to radiological findings, vascularity assessment of metastatic spinal tumors is mainly based on the histological tumor type [20,21]. Although tumor type is an indicator that can be easily assessed, variability in evaluation can still be observed, even within the same tumor type [21]. Our study shows that IFV diameter is a more reliable predictor for IBL than the histological tumor type, which may be due to the variability in evaluation within the same tumor type. Similar trends were observed in previous studies. Vascular assessments using CT-DSA and TR-CE-MRA showed discrepancies in the tumor type classification [31,32]. RCC, thyroid carcinoma, and HCC are generally classified as highly vascular cancers [20,21]. Tan et al. reported that approximately 67% of RCCs and 60% of HCCs appear to be hypervascular on angiography [33]. Inaccurate preoperative vascularity assessment can result in patients with hypervascular tumors missing essential preoperative embolization, which may lead to significant intraoperative bleeding. Given the potential for misclassification in common histological tumor type-based vascularity evaluations, radiological findings, including IFV diameter, should be considered during preoperative planning.

The frequency of IFV in metastatic spinal tumors has not yet been documented. When limited to metastatic bone tumors, several reports have suggested that flow voids are observed in approximately 46–94% of cases of RCC [12,13,14,34]. Murphy et al. showed that flow voids were most frequently identified in patients with RCC metastases (90%), followed by those with thyroid carcinoma metastases (60%) [14]. In our study, IFV was observed in 79.5% of the cases. Our findings suggest that IFV is a common observation not only in RCC but also across a variety of metastatic spinal tumor types. The frequency and characteristics of IFV in individual metastatic spinal tumor types remain subjects for future research.

Preoperative embolization (PE) reduces IBL during decompressive surgery for hypervascular spinal metastases [10,11]. The procedure is generally considered safe, with a risk of paralysis due to the embolization of the Adamkiewicz artery [35]. A meta-analysis of PE for spinal tumors reported an overall complication rate of 3.1%, with 16.2% of these complications resulting in permanent neurological deficits [36]. Therefore, an accurate preoperative vascularity assessment is crucial to avoid unnecessary PE. The efficacy of PE for hypervascular metastatic spinal tumors, as determined by the angiographic grade, has also been reported [21]. Given that angiography is invasive, future research should focus on the relationship between IFV diameter and angiographic grade, as well as the determination of appropriate cut-off values to minimize the need for invasive procedures.

This study has some limitations. First, as this was a retrospective study conducted at a single institution, the generalizability of the results is limited. Second, this study evaluated a variety of tumor types, which may have introduced variability in the results. Finally, owing to the retrospective nature of the study, selection and information biases may have been introduced. Future studies should aim to validate these findings through larger multicenter collaborative studies and explore the utility of IFV in specific cancer types. Prospective studies with standardized protocols are required to eliminate bias and provide more robust evidence. Additionally, increasing the sample size and focusing on single cancer types will help further elucidate the importance of IFV in surgical planning.

## 5. Conclusions

The IFV diameter was significantly associated with increased IBL during palliative excisional surgery for metastatic spinal tumors. A large IFV diameter indicates hypervascular tumors with intratumoral blood vessels that are challenging to control in terms of bleeding during excision. This readily assessable marker can help spine surgeons identify patients with a higher likelihood of massive bleeding. Future prospective multicenter studies with larger sample sizes and standardized protocols are warranted to validate these findings and explore the utility of IFV diameter as a reliable marker of vascularity across various tumor types in metastatic spinal tumors.

## Figures and Tables

**Figure 1 cancers-16-04124-f001:**
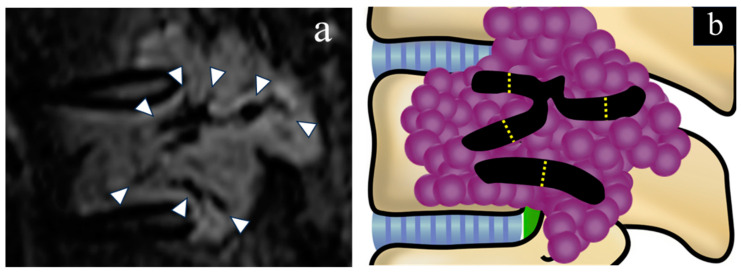
Schematic representation of intratumoral flow void (IFV) with standard magnetic resonance imaging (MRI). MRI images were acquired in three planes using T1-weighted, T2-weighted, and short-tau inversion recovery (STIR) sequences. (**a**) White arrows indicate IFV in a metastatic spinal tumor on sagittal STIR images. (**b**) The schematic diagram illustrates the intratumoral flow void diameter, which is defined as the greatest vertical distance across its tubular structure (yellow dotted line: vertical distance of IFV).

**Figure 2 cancers-16-04124-f002:**
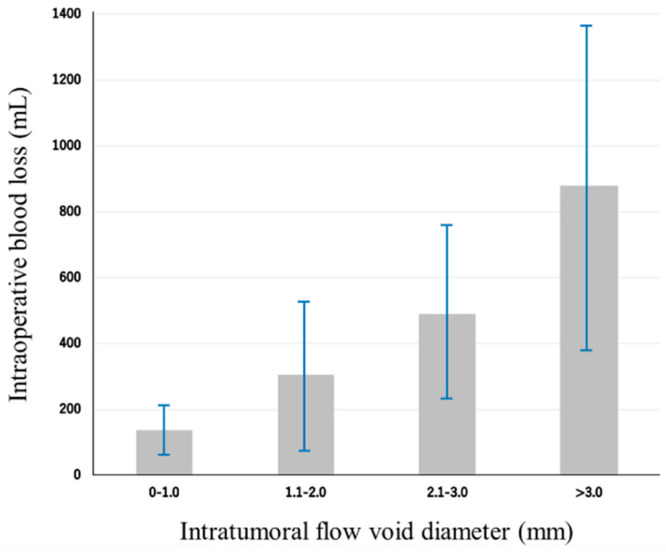
Distribution of intraoperative blood loss (IBL) by intratumoral flow void diameter. The mean IBL was 408 ± 382 mL. The mean intratumoral flow void diameter was 1.8 ± 1.3 mm. The amount of IBL increased progressively when we segmented the diameter of the flow void at 1 mm intervals. The average IBL for each category of intratumoral flow void diameter was as follows: 138 ± 72 mL for 0–1.0 mm (20 patients, 22.7%), 305 ± 236 mL for 1.1–2.0 mm (33 patients, 37.5%), 490 ± 277 mL for 2.1–3.0 mm (17 patients, 19.3%), and 878 ± 488 mL for >3.0 mm (18 patients, 20.5%).

**Table 1 cancers-16-04124-t001:** Demographic, oncological, surgical, and radiological characteristics.

Variables	Value
Mean age (years, mean ± SD)	61.6 ± 13.7
Sex (male)	56 (63.6)
BMI (kg/m^2^, mean ± SD)	23.1 ± 16.2
Tumor location	
Cervical	4 (4.5)
Thoracic	68 (77.2)
Lumbar	16 (18.2)
Tumor type (hypervascular)	19 (21.6)
SINS classification	
Stable	10 (11.4)
Potentially unstable	63 (71.6)
Unstable	15 (17.0)
Revised Tokuhashi score	
0–8 points	54 (61.3)
9–11 points	27 (30.7)
12–15 points	7 (8.0)
Preop. radiotherapy	11 (12.5)
No. of instrumented vertebrae	3.2 ± 1.0
No. of resected vertebrae	4.9 ± 1.9
Surgical duration (min)	237 ± 64
Presence of IFV	71 (80.7)
IFV diameter (mm)	1.8 ± 1.3

Values are presented as the number (%) of patients unless otherwise indicated. Abbreviations: BMI, body mass index; SINS, spinal instability neoplastic score; Preop., preoperative; No., number of; IFV, intratumoral flow void.

**Table 2 cancers-16-04124-t002:** Univariate analysis for intraoperative blood loss.

Variables	Correlation Coefficient	*p*-Value
Age (years)	0.04	0.72
Sex (male)	(−)	0.15
BMI (kg/m^2^)	−0.21	0.84
Tumor location (cervical, thoracic, lumbar)	(−)	0.17
Tumor type (hypervascular)	(−)	**<0.01**
SINS classification (stable, potentially unstable, unstable)	(−)	0.33
Revised Tokuhashi score (0–8, 9–11, 12–15)	(−)	0.85
Preop. radiotherapy	(−)	0.57
No. of instrumented vertebrae	0.39	**<0.01**
No. of resected vertebrae	0.20	0.06
Surgical duration (min)	0.44	**<0.01**
Presence of IFV	(−)	**<0.01**
IFV diameter (mm)	0.74	**<0.01**

Continuous variables were analyzed using Spearman’s correlation analysis; binary variables were analyzed using the Mann–Whitney U test; and categorical variables with three levels were analyzed using the Kruskal–Wallis test. Significant *p*-values (*p* < 0.05) are marked in bold. Abbreviations: BMI, body mass index; SINS, spinal instability neoplastic score; Preop., preoperative; No., number of; IFV, intratumoral flow void.

**Table 3 cancers-16-04124-t003:** Multivariate linear regression analysis for intraoperative blood loss.

Variables	Unstandardized Coefficient	Standardized Coefficient	95% CI	*p*-Value
Constant	−272	(-)	−509 to −35	
Tumor type (hypervascular)	77.2	0.083	−72 to 226	0.30
No. of instrumented vertebrae	−2.65	−0.13	−37 to 31	0.87
Surgical duration (min)	1.60	0.27	0.5 to 2.6	**<0.01**
IFV diameter (mm)	171	0.56	118 to 224	**<0.01**

Significant *p*-values (*p* < 0.05) with 95% CI are marked in bold. Abbreviations: CI, confidence interval; No., number of; IFV, intratumoral flow void.

## Data Availability

The data that support the findings of this study are available on request from the corresponding author. The data are not publicly available due to privacy or ethical restrictions.

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
