# Peer review of "Intratumoral Flow Void Diameter as a Predictor of High Intraoperative Blood Loss in Palliative Excisional Surgery for Metastatic Spinal Tumors"

_cancers, 2024, doi:10.3390/cancers16244124_

Round 1

Reviewer 1 Report

Comments and Suggestions for Authors

The Authors present a retrospective analysis regarding 93 cases of spinal metastasis undergoing palliative treatment in which the pre-operative flow-void signal was measures as possible predictors for intraoperative bleeding. The purpose of the study is interesting, however there are some issues that must be solved regarding mainly the methodology of this study.

Major issues:

-       There are no data regarding the application of the most common scores used for treatment algorithm of spinal metastases, which are SINS score and Tokuhashi score. Since palliative surgery has specific indications according to these scores, Authors should revise their dataset accordingly.

-       As the Authors states multiple time, pre-operative embolization is a non-invasive and safe procedure that allows to minimize the risk of intraoperative bleeding, however in this case series it has been performed only 5 times. Why?

Minor issues:

-       There are some minor grammars and synthax mistakes that should be corrected, like

“Surgical magnitude” should be “surgical site magnitude”

Comments on the Quality of English Language

 There are some minor grammars and synthax mistakes that should be corrected, like

“Surgical magnitude” should be “surgical site magnitude”

Reviewer 2 Report

Comments and Suggestions for Authors

The authors present a very interesting retrospective clinical study which identifies  IFV diameter in standard diagnostic MRIs an independent predictor of intraoperative blood loss in patients who underwent palliative surgery fpr metastatic spinal tumors. The IFV diameter can be easily assessed using standard MRI.  93 surgical cases where included in this study. 

My main concern is the fact that 5 patients in this surgical series received preop. tumor embolization. I assume that IFV or IFV diameters in MRI scans of these patients  where assessed before embo and not afterwards. If so, these patients need to be excluded from the analysis. 

Reviewer 3 Report

Comments and Suggestions for Authors

In this study, numerous patients information were collected to answer the hypothesis of intratumoral flow effect on intraoperative blood loss  in palliative excisional surgery. However, there are some comments for authors to be further clarified.

1. The article was entitled as "Impact of Intratumoral Flow Void on Intraoperative Blood Loss in Palliative Excisional Surgery for Metastatic Spinal Tumors". Can authors highlight or demonstrate more clear to indicste what kind of impact they did? Otherwise, there were too much possible impacts occurred that associated with  Intraoperative Blood Loss in Palliative Excisional Surgery.

From the experimental layout, it was difficult to clarify the association of  Intratumoral Flow Void and Intraoperative Blood Loss in Palliative Excisional Surgery, therefore, the article presented weak scientific significance. Please provide more data or information to show the intratumoral flow void is critical issue in blood loss in palliative excisional surgery.

3. In terms of table, authors provided several tables in thi sstudy, however, these tables cannot help us to understand the results or associations in authors hypothesis. The suggestion is that can these data from tables be swtiched into figures which can help us to understand the results  or associations in the hypothesis.

Round 2

Reviewer 3 Report

Comments and Suggestions for Authors

Authors  did well revesion for this article and make it much scientific significance. Here, only one suggestion for authors, 

Please remove any additional lines or words from figure 2 to make it clearer.